# MANIPULATING MULTI-AGENT NAVIGATION TASK VIA EMERGENT COMMUNICATIONS

## ABSTRACT

Multi-agent corporations struggle to efficiently sustain grounded communications with specific task goal. Existing approaches are limited in their simple task settings and single-turn communications. This work describes a multi-agent communication scenario via emergent language in navigation task. This task involves two agents with unequal abilities: the tourist (agent A) who can only observe its surroundings and the guide (agent B) who has the holistic view but does not know the initial position of agent A. They communicate with the emerged language grounded through the environment and a common task goal: help the tourist find the target place. We release a new dataset of 3000 scenarios that involve such visual and language navigation. We also seek to address the multi-agent emergent communications by proposing a collaborative learning framework that enables the agents to generate and understand emergent language and solve task. The framework is trained with reinforcement learning by maximizing the task success rate in an end-to-end manner. Results show that the proposed framework achieves competing performance in both accuracy of language understanding and in task success rate. We also discuss the explanations of the emerged language.

## 1 INTRODUCTION

Communication is a crucial factor for multiple agents to cooperate. While most recent works are focused on the interactions between the artificial agent with humans, there still are some research efforts made on communication between artificial agents. However, most of these works are focused on single-turn communication with a unidirectional message pass and the evolvement of natural language or some specific properties of the emergent language like compositionality, interpretability, and so on. But the multi-turn conversation is more analogous to human language. In a natural conversation, language generation and understanding should be mutual rather than unidirectional. Therefore, we provide a framework for agents to generate multi-turn dialogues including two agents.

To prove the feasibility of our framework, we provide a scenario for agents to generate multi-turn conversations. We propose a new task adapted from the vision-language navigation(VLN) task coming from the human-machine communication area where the guide should communicate with the tourist to give guidance and help it find the target location. Different from traditional VLN tasks, in our settings, the tourist(agent A) and the guide(agent B) are both machines, rather than the original human-machine settings. Besides, we suppose that the guide does not know the initial position of the tourist, so the guide not only has to give guidance to the tourist but also confirm the location of it.

Our contributions can be summarized as follows:

1. From the view of emergent language, we study the language with multiple turns. To give a suitable scenario for multi-turn conversations, we provide a navigation task adapted from the vision-language navigation(VLN) task.

2. Compared with methods with agents speaking the natural language, ours is cheaper with no expensive annotation, making it a more practical way for agents to communicate.

3. As far as we know, We are the first to propose a VLN-like task in a two-agent cooperation scenario. And we also provide a benchmark for it, which gives a possible solution to this kind of task.

## 2 RELATED WORK

**Vision Language Navigation(VLN)** Navigation tasks with vision and natural language information have attracted much attention in the last few years. Generally, this task involves humans, an artificial agent, and the environment. The agent communicates with humans in natural language. The agent may receive requests or instructions and ask for more guidance from the human. It can navigate and interact with the environment to get more information or complete corresponding task requested by the human. While humans should give requests or guidance to help the agent complete the task.

Generally, the goal of these VLN tasks is to help the agent find the target place known only by the human side. Anderson et al. (2017) proposes a VLN dataset called Room-to-Room (R2R) based on matterport3d Chang et al. (2017) data and provides a benchmark for this task first. Chen et al. (2018) proposes a dataset Touchdown and applies the VLN task to the outdoor environment. Still, some scenarios where agents need to interact with objects in the environment are proposed. Narayan-Chen et al. (2019) put the task into the Minecraft game and instruct the agent to complete building tasks.

From the view of language format, we can divide those tasks into three categories Gu et al. (2022):

- Initial instructions: The agent is given instructions for the whole navigation episode. In some of those works, the instructions are very detailed Anderson et al. (2017); Chang et al. (2017); Jain et al. (2019); Krantz et al. (2020). For example, "Walk down one flight of stairs and stop on the landing". While in other works, some coarse-grained instructions are given Zhu et al. (2021). However, this kind of work only gives the instructions at the beginning, so multi-turn communication can not be generated.

- Oracle guidance: In the process of navigation, the agent can request additional natural language guidance to get more information Chi et al. (2019). But in most cases, the oracle can only respond with simple words like "turn left" and "turn right".

- Dialogue: Dialogue is given as a supervised signal.de Vries et al. (2018); Thomason et al. (2019). But the fixed dialogue constrains the flexibility of policy.

By putting the VLN task into multi-agent settings, the problems listed above can be alleviated or solved. To our knowledge, talk-the-walk de Vries et al. (2018) is the only work applying the VLN task with multiple agents and is very similar to our task settings. However, talk-the-walk only focuses on the localization part, where the guide finds where the tourist is. But ignore the guidance part, in which the guide makes a route plan and gives guidance to the tourist, and the tourist makes corresponding actions. Talk-the-walk models this part with a random walk and only implements a minimal baseline, making the VLN task with multi-agent settings incomplete.

**Emergent Language** Emergent language is an unplanned language that comes up in interactions between the students, and/or the teacher and the students. From the perspective of agents' relationships, most of the works are focused on fully cooperative tasks like referential games Lazaridou et al. (2018) adapted from the Lewis signaling game. In the referential game, there are two roles: speaker and listener. The speaker has to give instructions to the listener so that it can select the target picture from the candidates successfully. Some navigation tasks are proposed Kalinowska et al. (2022); Kajić et al. (2020), however, those works can only be applied in limited environments, with 2 and 4 cases respectively. Some semi-cooperative tasks are also proposed, like the negotiation game Cao et al. (2018), in which two agents have to negotiate to get more scores for themselves. There are also some fully-competitive cases like a circular-based sender-receiver game Noukhovitch et al. (2021) where the sender tries to make the receiver choose the position closer to its own target position rather than the receiver's. Most of those part of works are studying single message pass, with one agent only having a language generation part, and the other one having a language understanding module. Still, there are some works putting attention to the multi-turn form Cao et al. (2018); Evtimova et al. (2017). But the former study focuses on a semi-cooperative case rather than a fully cooperative one like us, while the latter one uses the bag of words, and generates 0 or 1, with 1 indicating this token is in the generated sentence. This method makes that the latter token does not rely on the former ones, making it not a sentence from general views.

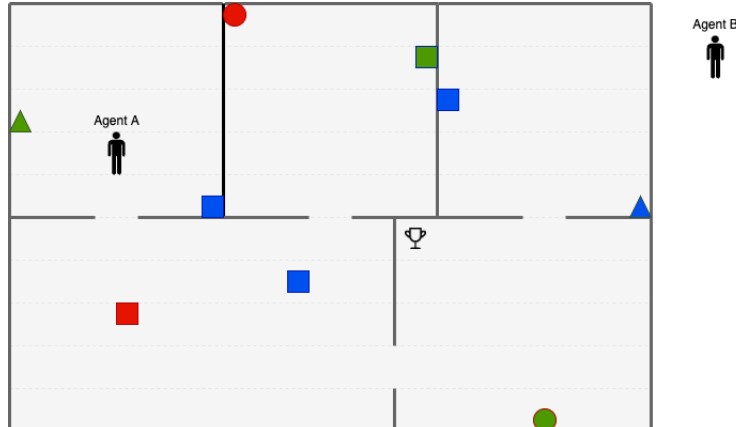

Figure 1: The navigation task involves three components: the tourist (agent A), the guide (agent B), and the environment. Agent A and B have unequal abilities. While agent A can only see its surroundings and describe them to B, agent B has the global perspective, can make a route plan, and give instructions to A. Agent A and B cooperate to lead A to the target place.

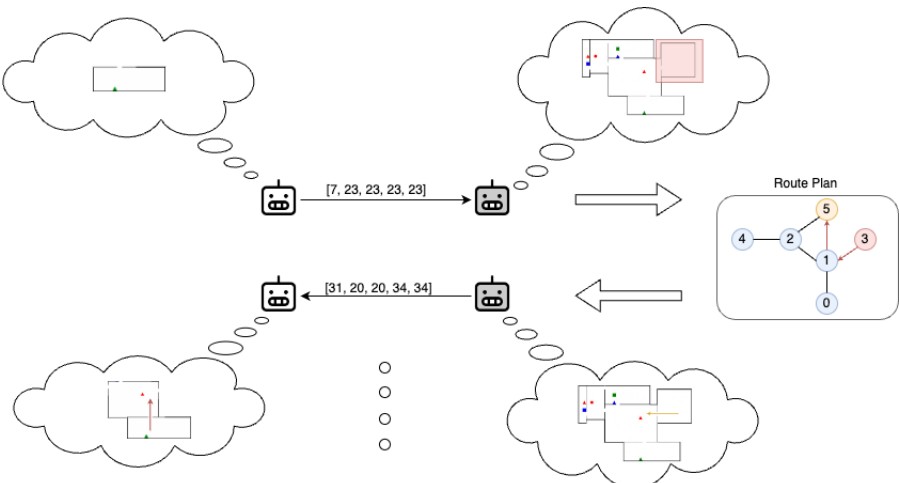

Figure 2: Navigation Case. The navigation task consists of three sub-tasks: Localization, Route Plan, and Guidance. At every turn, agent A first gives a description of its surroundings in the room it is located. With the current messages and the history messages given by A, agent B tries to guess where A is using the topological structure of the rooms. Then, based on this guess, agent B makes a route plan for A to get to the target place, and gives an instruction to A describing the next step A should take.

## 3 NAVIGATION TASK

This task is inspired by a real-world scenario where two agents with unequal abilities, agents A and B, cooperate to help agent A find the target place. As pictured in Figure 1, agent A can only see the objects around it, while agent B has the floor plan and the target place but doesn't know where A is. To win this game, they have to cooperate to help A reach the target place with limited rounds. Otherwise, this task fails.

Here, we divide the navigation task into three steps: First, localization. Agent B needs to find where A is, according to the message given by A describing the objects around it. Second, route plan. Having the position of A and the target place, agent B has to find the shortest path for A to get there. Third, guidance. Agent B gives instructions to A, helping it follow the route plan. All three steps above are executed iteratively until A reaches the target place or the steps run out.

Formally, this navigation task is constructed as follows:

There are houses, containing rooms $R = \{r_0, r_1, ..., r_N\}$ each. Every time, one house is randomly chosen from those $K$ candidates. Then two rooms are drawn from the selected house, and one of them is the target room $r_o$, and the other is the start room $r_s$. Every room has several gates $D_i = \{d_{ij}, j \in Neighbor(i)\}$. There is a graph for each house, to model the relationship between rooms. We use $G = (R, D)$ to denote a house, where $R$ is the vertex set, and $D$ is the edge set. There are two players called agent A, and agent B parameterized by $\theta_A, \theta_B$ separately. The inputs and outputs are different with different tasks.

- In the localization task, agent A takes in the room's vector where it is and gives out the messages of step t consisting of $M$ tokens $m_t = \{v_0, v_1, ..., v_M\}, v_i \in V$. While agent B collects history dialogue $h_t = (m_0, m_1, .., m_t)$ and tries to guess where A is.

- In the route plan task, agent B uses the room graph $G$, the guessed position of A $\arg\max(\theta_B(h_t, G))$, and the target position $r_o$ to get the next step A should take $\arg\max(\theta_A(h_{t+1}, D))$.

- The only difference between guidance and localization task is that the guidance task switches the role of agents A and B, and agent A does not require the graph $G$.

A case is shown in Figure 2, where Agent A starts from room 0 and wants to get to room 5.

## 4 MODEL

Figure 3 shows the model structure for the localization task. This model includes an encoder encoding the surroundings of A, a route encoder embedding the neighbor information by sampling routes ended with the target room, a language generation module in agent A implemented with GRU, and a language understanding module in agent B used for guessing the position of A.

Firstly, the information about the house containing features of a set of rooms $(r_0, r_1, .., r_N)$ is given. Agent A encodes the target room information $r_o$ including objects and gates and then uses the encoded vector to generate corresponding emergent language with its language generation module $e_r^o = GRU(MLP(r_o))$. With the message coming from agent A in this turn, and the history sentences from A, agent B makes its guess by making use of the topological structure of rooms $\theta_B(h_t, G)$. In this topological graph, every room is represented with a node, and every door connecting neighbor rooms is represented with an edge between these rooms. For node $k$, we sample $T$ routes with the same length of history movement length $t$ ended with this node $route_j^k$. Node $k$ has representation $e_s^k = \frac{1}{T} \sum_i^T GRU(route_i^k)$ including topological structure information.

And B encodes the history sentences with another GRU and computes the similarity between this encoded history $e_{m_t}$ with each node representation $e_s^i$ calculated above to find the most possible position of agent A. Similarly, the model for the guidance task also includes an encoder, a language generation, and a language understanding module used for giving instructions to agent A about its next step.

Considering the non-differentiability of the discrete messages passing between agents, those models are trained with REINFORCE algorithm. We use the Floyd algorithm for the route plan task to find

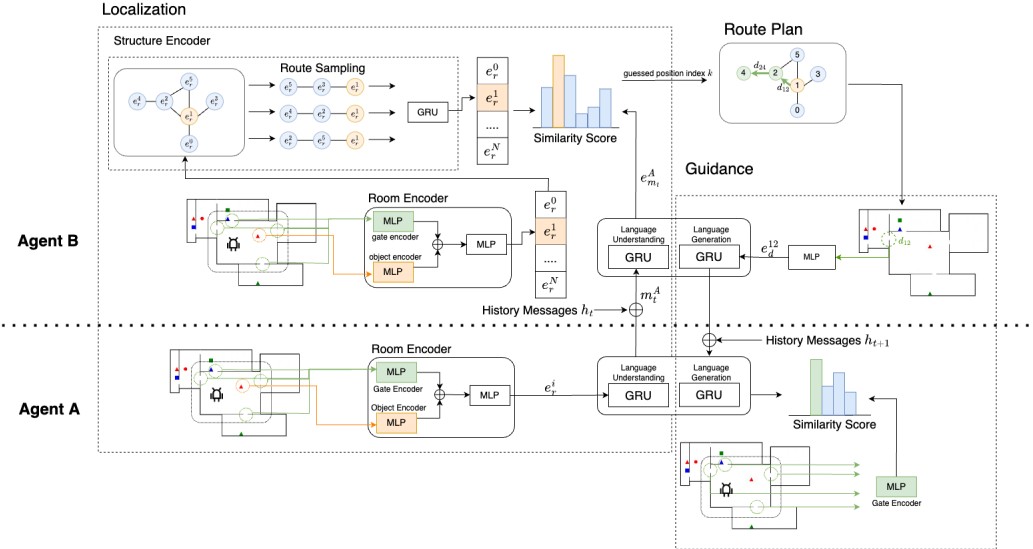

Figure 3: Model Structure. There are three sub-modules here, including localization, route plan, and guidance. In the localization part, by calculating the similarity score between messages given by A and room feature embeddings incorporating structural information, agent B tries to find where A is. Then we use the Floyd algorithm to find the shortest path between the guessed place and the target one. And in the guidance module, A could find its path with history dialogue with B and its own observations.

the shortest paths. Losses in localization and guidance tasks can be calculated with the following equations:

$$
\begin{aligned}
\bigtriangledown_{\theta_A} J_{localization} &= E[\frac{1}{M}R(\cdot)\sum_i^M \bigtriangledown_{\theta_A} \log \pi_A(v_i|s_A)] \\
\bigtriangledown_{\theta_B} J_{localization} &= E[R(\cdot)\bigtriangledown_{\theta_B} \log \pi_B(r|s_B)] \\
\bigtriangledown_{\theta_B} J_{guidance} &= E[\frac{1}{M}R(\cdot)\sum_i^M \bigtriangledown_{\theta_B} \log \pi_B(v_i|s_B)] \\
\bigtriangledown_{\theta_A} J_{guidance} &= E[R(\cdot)\bigtriangledown_{\theta_A} \log \pi_A(d|s_A)]
\end{aligned}
\tag{1}
$$

Finally, we put those sub-tasks together to complete the navigation task. In the training process, the input of the localization part is the surroundings of agent A, and the output of it is the guessing position made by agent B. However, the input of the route plan part in the next timestep is the actual position of A, instead of the guessing one, preventing the error from localization influencing other modules. While in the testing process, the input of the route plan module comes from the last localization module.

## 5 EXPERIMENT

### 5.1 DATASET

We construct the dataset with 6 rooms for each navigation task. And every object in this dataset has 3 optional colors and 3 optional shapes. Also, some features not relevant to this task are added, like the gate color which we suppose to be the same among all rooms in the same house.

### 5.2 DISCRETE VAE

Rolfe (2016) proposes discrete VAE. By solving the problem of backpropagation through discrete variables, discrete VAE successfully constructs the semantic representation of pictures.

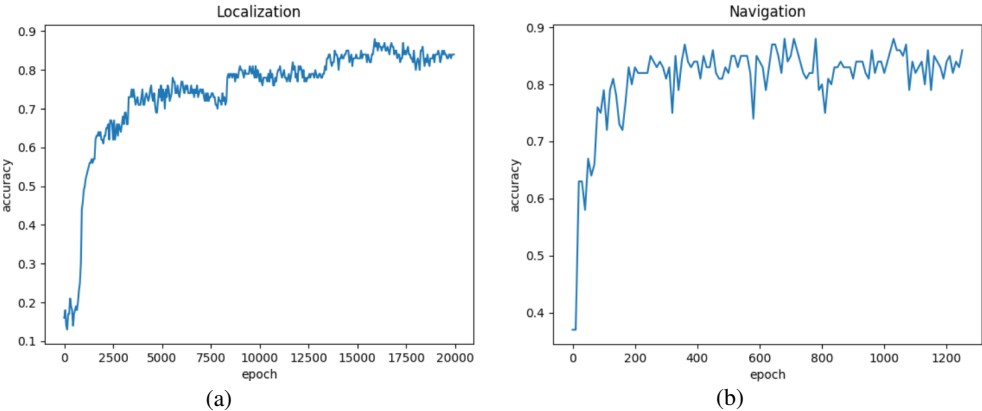

(a)                (b)

Figure 4: The change of accuracy over epoch in different tasks. (a) localization task. (b) navigation task

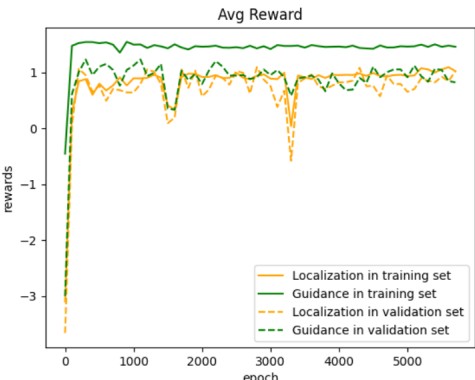

Figure 5: The reward changes of the training set and the validation set. The rewards tend to be stable over epochs, proving the effectiveness of our framework.

We apply the discrete VAE as a comparable model here by replacing the room encoder and gate encoder together with the language generation part. Fixing the discrete VAE part, train other modules, then we get a comparable model.

## 5.3 RESULTS

Figure 4 and Figure 5 show the accuracy change in the validation set and the reward change over the epoch separately. In this experiment, the length of messages passing between agents A and B is 3, and there are 20 candidate tokens in the localization task and 40 candidate tokens in the navigation task. It is worth noting that, since it is trained from scratch, there is no initial meaning for those candidate tokens.

Table 1 gives some comparisons between the model adapted from discrete VAE and our model. It should be noted that the adapted discrete VAE already has the initial meaning of each token before training on the specific tasks. Considering that the presentations of room and routes are trained based on the reconstruction task in the discrete VAE model, making the discrete presentations contain more redundant information. With the limitations of the message length and vocabulary size in discrete communication scenarios, discrete VAE may communicate less useful information compared with our model. We also show the case when agents can pass continuous messages, giving us a ceiling performance when the communication channel is not restricted.

| Model | Task | Accuracy | $\rho$ | purity |
|-------|------|----------|--------|--------|
| continuous | Localization | 0.96 | - | - |
| continuous | Navigation | 0.89 | - | - |
| discrete VAE | Localization | 0.81 | -0.0113 | **0.556 / 0.667** / 0.148 |
| ours | Localization | **0.84** | **0.4775** | 0.407 / 0.457 / **0.235** |
| discrete VAE | Navigation | 0.69 | - | - |
| ours | Navigation | **0.86** | - | - |

Table 1: Comparison between adapt discrete VAE and our model. $\rho$ is the topographic similarity, the purity scores are calculated when labeled with object color/shape/direction

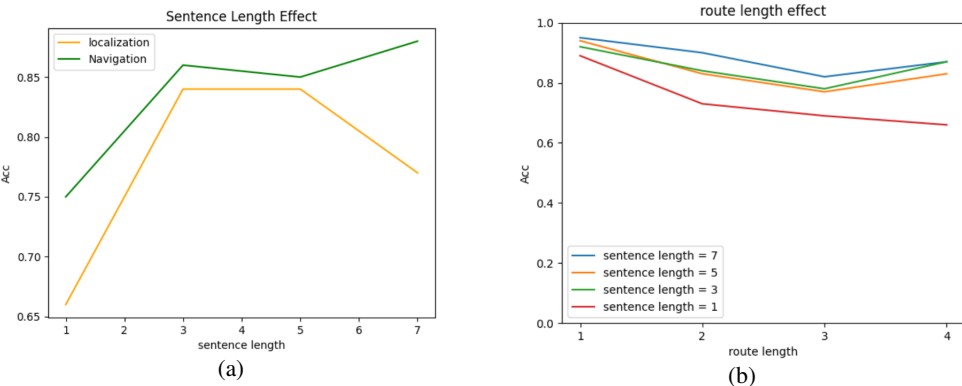

(a)  (b)

Figure 6: The effect of (a) sentence length and (b) the shortest route length between the start and target places on the task performance

To evaluate the generated language, we use topographic similarity and purity. We suppose that good language tends to give similar expressions describing a similar object. So we use topographic similarity $\rho$ as our metric, which is the Spearman correlation between the message similarity sequence and the room vector similarity sequence. And our method outperforms the discrete VAE largely. To make the emergent language more interpretable and give some more hints about the information it encoded, we calculate the purity score with the following procedure: First, we only change the color, shape, and direction attributes of the object and fix other features. Then, we cluster those rooms with the generated message. Labeled these rooms with their objects' color, shape and direction separately, we get the corresponding purity. And the purity score can tell us the information that the emergent language encoded. From the table, we can tell that compared with discrete VAE, our model focus more on the direction and less on color and shape which is reasonable because there are only 3 choices for color and shape which makes it less useful for disguising the target room from others, comparing with the direction with 8 possible value.

We also investigate factors that may affect task performance 6. Countering with our intuitions that, as the sentence length grows, the model has a much more powerful expression ability, so the performance should grow, but localization accuracy grows first and then drop. When the sentence length is enough for expression, much more tokens may increase the difficulty for the model to tackle. The shortest route length between the start positions and the target ones can also affect the task performance, but it does not follow an exponential drop when the shortest route length grows.

## 6 CONCLUSION

In this work, we propose a navigation task in an emergent communication scenario. This task is adapted from the vision-language navigation task proposed in human-machine communication. However, compared with the traditional VLN task, the application of multi-agent communication in the VLN task makes the communication policy more flexible and can generate a more complex

dialogue format. Our method proves the feasibility of applying multi-agent communication to the VLN task, and also gives a benchmark for this task.

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
