# OpenReview forum: "Manipulating Multi-agent Navigation Task via Emergent Communications"
_ICLR.cc/2023/Conference — Submitted to ICLR 2023_

### Official Review · Reviewer_SZp3 · 2022-10-23

**Confidence:** 5
**Correctness:** 2
**Technical Novelty And Significance:** 2
**Empirical Novelty And Significance:** 2
**Recommendation:** 1

**Clarity, Quality, Novelty And Reproducibility:**

Overall, the quality of the paper is very poor. The paper only has 7 pages but leaves out a lot of important details. The related work misses a lot of important related on dialog-based vision-language navigation [1, 2, 3, 4, 5]. I suggest the authors conduct a more thorough literature search in this space.

[1] https://arxiv.org/pdf/1909.01871.pdf

[2] https://arxiv.org/pdf/1812.04155.pdf

[3] https://arxiv.org/pdf/1907.04957.pdf

[4] https://arxiv.org/pdf/2110.00534.pdf

[5] https://aclanthology.org/P19-1537.pdf

**Strength And Weaknesses:**

Strengths:
* The proposed setting is novel.

Weaknesses:
* While novel, the proposed setting is poorly motivated. Why studying this setting is significant? What would it enable?
* The description of the environment is very nebulous. Especially, the complexity of the visual and language components are highly under-specified. How is the input observation represented (image or feature-based)? How many object categories? How large is the agent's vocabulary? How are the environment graph generated? It is thus difficult to comprehend the significance of solving this environment.
* The plots are missing confidence intervals. Each experiments are not conducted multiple times with different random seeds. This seriously weakens the empirical claims.

**Summary Of The Paper:**

The paper proposes a multi-turn, bidirectional emergent communication setting implemented as a vision-language navigation task. Experiments show that the trained agent achieves high accuracy in localization and navigation tasks. The proposed model outperforms a VAE baseline.

**Summary Of The Review:**

While the idea is interesting, given its current poor writing and execution quality, I recommend rejecting the paper.

---

### Official Review · Reviewer_3NDJ · 2022-10-24

**Confidence:** 5
**Correctness:** 2
**Technical Novelty And Significance:** 2
**Empirical Novelty And Significance:** 1
**Recommendation:** 1

**Clarity, Quality, Novelty And Reproducibility:**

The paper is well-written.
The novelty is not very high as it is a simplification of "Talk the Walk", or an improvement over Kajic2021. Yet, if correctly explored, it is still valuable

**Strength And Weaknesses:**

While this paper goes in a good direction by moving away from the Lewis game in the language emergence communication setting, it falls short as there is little scientific analysis. The authors mainly designed an artificial navigation task, handcrafted an observation space and reward, designed a neural network, trained it, and reported the results. Those results are not put into perspective with a detailed researched perspective. Indeed, the point of Emecom in ML is not to create artificial tasks for the sake of solving it, but to design tasks that could help understand how language evolves with a human intervention as possible (or being generic).

I am listing here a few leads to improve the paper in the long run:
 - The introduction should frame the research problem into the past literature, ask a scientific question, and explain how the paper help answers it. Here, there is no citation; the VLM task is only said to be more natural (no reference to the embodiment, situated learning, or going beyond the signaling game)
 - The related work is extremely short. 18 citations in total for a paper about language emergence, and the navigation task is just inherently incomplete. On a positive note, the most relevant citations kajic/Kanilowska/DeVries are present
- The paper is 7 pages long, and page 3 is barely empty
- The model is designed to solve the task, e.g., the graph networks are specific to the observation space. I would recommend either picking a minimal architecture or reproducing the architecture that is classic in VLM.
- having 4 losses in an RL task is generally a bad sign as it is a symptom of human intervention. cf The bitter lesson by Sutton
 - Results are not compared to any baselines (VAE vs your network is not really a baseline...), and simply reporting the success rate is not sufficient. Examples of suitable ablation include: the correlation between language and environment complexity. Qualitative experiments to translate language and change in the environment (e.g., how changing the graph impacts the language), what are the external constraints that shape language, reward ablation, etc.

I would recommend the authors to re-work their paper and present it to a workshop to have further guidance. The initial idea is good, but the realization lacks scientific maturity.






**Summary Of The Paper:**

This paper explores language emergence in the context of a navigation/coordination task.To do so, they introduce a simple gridworld environment sympliying the orginal "Talk the Walk" task. The authors then propose one RL model to solve the task.

**Summary Of The Review:**

For all the reasons explained above, the paper is clearly below ICLR standard. I provided a few advise, that may help the authors improving

---

### Official Review · Reviewer_6nP4 · 2022-10-25

**Confidence:** 4
**Correctness:** 2
**Technical Novelty And Significance:** 1
**Empirical Novelty And Significance:** 2
**Recommendation:** 1

**Clarity, Quality, Novelty And Reproducibility:**

The work is not completely novel as there are other better frameworks present as highlighted above.

**Strength And Weaknesses:**

The paper is not clearly written and some sections are hard to understand. The related works do capture how the framework proposes new challenges that are not already captured in previous work. Besides a similar work that also uses communication between agents [2] perform a similar analysis and comparisons to that work would be help in highlighting the difference with the proposed framework.

Besides the scope of the proposed framework is limited as it can only handle artificial messages that have a much smaller length, vocabulary, and sources of variation. In contrast, previous work have already proposed similar benchmarks that involve real images with natural language descriptions [1].

[1] de Vries et al 2018. Talk the Walk: Navigating New York City through Grounded Dialogue.

[2] Singh et al. 2019. Learning when to Communicate at Scale in Multiagent Cooperative and Competitive Tasks.

**Summary Of The Paper:**

The paper proposes a multi-agent navigation framework as a new benchmark to assess the emergent language that is thus evolved between the agents.

**Summary Of The Review:**

The paper appears to be a work in progress without much emphasis put on literature survey and comparisons to prior work. The paper also misses out on performing extensive empirical evaluation of the current state-of-the-art methods used for multi-agent navigation tasks.

---

### Decision · Program_Chairs · 2023-01-20

**Decision:**

Reject

**Justification For Why Not Higher Score:**

The paper has several critical weaknesses as stated above. The reviewers were unanimous. Hence I am unable to provide a higher rating.

**Justification For Why Not Lower Score:**

N/A

**Metareview: Summary, Strengths And Weaknesses:**

This paper studies the emergence of language in the context of a vision and language navigation task. The paper introduces an interesting grid world environment with two agents: a tourist who is trying to find a target and a guide who has some privileged information but does not know the initial location of the agent. This is an interesting research topic and the presented environment has potential. However, all three reviewers agreed that the paper is not yet ready for publication at ICLR. A few notable weaknesses include the lack of comprehensive comparison to previous work, incomplete positioning of the work and environment in the context of a large body of related work and the writing. The reviewers were unanimous in their scores. No rebuttal was provided. Given the above I am recommending rejecting the paper.